# Light Regulation of Two New Manganese Peroxidase-Encoding Genes in *Trametes polyzona* KU-RNW027

**DOI:** 10.3390/microorganisms8060852

**Published:** 2020-06-05

**Authors:** Piyangkun Lueangjaroenkit, Emi Kunitake, Makiko Sakka, Tetsuya Kimura, Churapa Teerapatsakul, Kazuo Sakka, Lerluck Chitradon

**Affiliations:** 1Department of Microbiology, Faculty of Science, Kasetsart University, Bangkok 10900, Thailand; fscipkl@ku.ac.th (P.L.); fscicpt@ku.ac.th (C.T.); 2Laboratory of Applied Microbiology, Graduate School of Bioresources, Mie University, 1577 Kurimamachiya-cho, Tsu, Mie 514-8507, Japan; Kunitake@bio.mie-u.ac.jp (E.K.); makiko@bio.mie-u.ac.jp (M.S.); t-kimura@bio.mie-u.ac.jp (T.K.); sakka@bio.mie-u.ac.jp (K.S.)

**Keywords:** light, manganese peroxidase, *Trametes polyzona*, gene expression, enzyme production, light responsive element

## Abstract

To better understand the light regulation of ligninolytic systems in *Trametes polyzona* KU-RNW027, ligninolytic enzymes-encoding genes were identified and analyzed to determine their transcriptional regulatory elements. Elements of light regulation were investigated in submerged culture. Three ligninolytic enzyme-encoding genes, *mnp1*, *mnp2*, and *lac1*, were found. Cloning of the genes encoding MnP1 and MnP2 revealed distinct deduced amino acid sequences with 90% and 86% similarity to MnPs in *Lenzites gibbosa*, respectively. These were classified as new members of short-type hybrid MnPs in subfamily A.2 class II fungal secretion heme peroxidase. A light responsive element (LRE), composed of a 5′-CCRCCC-3′ motif in both *mnp* promoters, is reported. Light enhanced MnP activity 1.5 times but not laccase activity. The *mnp* gene expressions under light condition increased 6.5- and 3.8-fold, respectively. Regulation of laccase gene expression by light was inconsistent with the absence of LREs in their promoter. Blue light did not affect gene expressions but impacted their stability. Reductions of MnP and laccase production under blue light were observed. The details of the molecular mechanisms underlying enzyme production in this white-rot fungus provide useful knowledge for wood degradation relative to illumination condition. These novel observations demonstrate the potential of enhancing ligninolytic enzyme production by this fungus for applications with an eco-friendly approach to bioremediation.

## 1. Introduction

Light is an important environmental factor for phototrophic organisms, such as plants, algae, and phototrophic microorganisms, and for non-phototrophic microorganisms, including fungi [1]. Fungi use photoreceptors as eye-like sensors to detect light of specific wavelengths, such as phytochrome for red light, opsin for green light, white collar and vivid flavoprotein for blue light, and cryptochrome for white light [2,3,4,5]. Therefore, light-driven regulation corresponds to specific types of photoreceptors. Fundamental light-driven processes in fungi include cell development [6], protein secretion [7], conidiation [8], fungal spore germination [9], fruiting body formation [10], virulence of pathogenic fungi [11], carotenoid synthesis [12], and pigmentation [13]. 

The effect of light on the production of ligninolytic enzymes such as lignin peroxidase (LiP) and laccase has been investigated previously [14,15]. Green light enhanced lignin peroxidase production in *Phanerochaete chrysosporium* under submerged culture, whereas red, blue, and UV light reduced productivity [14]. Blue light efficiently boosted laccase production in *Cerrena unicolor* and *Pycnoporus sanguineus*, whereas white light increased laccase production in *Phlebia lindtneri* [15]. These results indicated that light diversely affects each fungus. Light-dependent control of transcription using an artificial on-off switchable promoter in *Trichoderma reesei* was reported [16]. 

The *cis*-acting regulatory sequence elements are contained in promotor regions that are selectively recognized by *trans*-acting transcription factors to regulate gene expression [17]. Several functional *cis*-acting regulatory sequence elements have been identified in promoter regions to achieve activation or deactivation of mRNA translation. Metal responsive element (MRE) and xenobiotic responsive element (XRE) are commonly-found *cis*-acting regulatory sequence elements which are involved in non-specific metal-ion and aromatic-compound-regulated gene expression of ligninolytic enzymes [17,18,19]. Light responsive elements (LREs) have been identified in plant promoters for light-regulated gene expression systems [20]. However, no reports have been published concerning the effects of LREs on promoters of ligninolytic enzyme-encoding genes.

*Trametes polyzona* KU-RNW027 is a white rot fungus that exhibits high manganese peroxidase (MnP) and laccase activities with efficacy on dyes and pharmaceutical product degradation [21,22]. Here, the aim was to investigate the enzyme-encoding genes in the ligninolytic system of *Trametes polyzona* KU-RNW027 and how they regulate enzyme production and enzyme-gene expression under conditions of illumination. Genes encoding two MnPs and one laccase were cloned and identified. Phylogenetic relationships of MnPs and laccase were analyzed by their amino acid sequence similarities. The promoter sequences of *mnp1*, *mnp2*, and *lac1* genes were analyzed for *cis*-acting regulatory sequence elements. The light-driven regulation of ligninolytic enzyme production and its gene expressions in a submerged culture of *T. polyzona* KU-RNW027 were also investigated.

## 2. Materials and Methods

### 2.1. Microorganism and Inoculum Preparation 

The white rot fungus, *T. polyzona* KU-RNW027, was maintained on potato dextrose agar (PDA) and stored in 20% glycerol at –20 °C for long-term preservation. For inoculum preparation, this fungus was cultured in 50 mL Kirk’s liquid medium for 3 days to prepare fungal pellet inoculum following Lueangjaroenkit et al. [21]. Fungal pellets (0.05 g/L) were used as inoculum in all experiments.

### 2.2. Plate Assay of Light-Driven Ligninolytic Enzyme Activity

A plug of mycelial tip with a 0.5 cm diameter was inoculated on PDA plates supplemented with 1 mM 2,6-dimethoxy phenol (2,6-DMP). The fungus was cultured under continuous lighting with a tungsten incandescent bulb. Light intensity was set to 500 lux. For the dark control, plates were fully covered with aluminum foil. PDA without 2,6-DMP was used as the control. All cultures were incubated at 30 °C for 3 days. Ligninolytic enzymes oxidized 2,6-DMP to an orange product, allowing the visualization of enzyme activity.

### 2.3. Effect of Light on Transcription Levels and Enzyme Production

*T. polyzona* KU-RNW027 was grown in 50 mL of Kirk’s liquid medium and exposed to light sources for 7 days. Tungsten incandescent bulb and blue light emitting diode (LED) were used. Tungsten incandescent bulbs had a broad spectrum of 320–1100 nm. The wavelength of the blue LED was 465 nm and the light intensity was set to 500 lux. The cultures were incubated under shaking at 130 rpm, 30 °C. The control was incubated in the dark and flasks were fully covered with aluminum foil to prevent light penetration. Ligninolytic enzyme activities, gene expression at the transcription level, and fungal growth were determined. All experiments were performed in triplicate. 

### 2.4. Enzyme Stability under Different Wavelengths of Light

Enzyme stability toward different light wavelengths was evaluated at 30 °C by pre-incubating the ligninolytic enzyme under continuous lighting with a tungsten incandescent bulb for visible light and blue LED light for 72 h. Light intensity was set to 500 lux. Residual activities were determined at 1, 3, 6, 24, 48, and 72 h. All experiments were performed in triplicate. 

### 2.5. Enzyme Assay

MnP and laccase activity assays were performed according to the method of Lueangjaroenkit et al. [21] and Kondo et al. [23]. MnP and laccase activities were determined by monitoring the oxidation of 2,6-DMP at 469 nm, and its absorbency change followed within 3 min. For laccase assay, the reaction mixture consisted of 1 mM 2,6-DMP in 50 mM malonate buffer (pH 4.5). For the MnP assay, the reaction mixture consisted of 1 mM 2,6-DMP, 1 mM MnSO_4_, and 0.1 mM H_2_O_2_ in malonate buffer (pH 4.5). The result for the latter procedure is expressed as total activity (laccase + MnP activity). Differences between activities of laccase and total activity were used to calculate MnP activity. Enzyme denaturation at 100°C for 15 min was performed as a control reaction in parallel. One unit (U) of either MnP or laccase was defined as activity oxidizing 1 µmol of substrate per min.

### 2.6. Cloning of Ligninolytic Enzyme-Encoding Genes and Promoter Region 

Genomic DNA was extracted from the mycelia cultivated in Kirk’s liquid medium as previously described [21]. In brief, mycelia were ground in liquid nitrogen and lysis extraction buffer was then added. After incubation on ice for 30 min, 50 µL of 5 M potassium acetate (pH 4.8) was added and the mixture was further incubated on ice for 15 min before centrifuging. The supernatant was transferred into a new tube. DNA precipitation was conducted using isopropanol. Genomic DNA was washed with 70% ethanol. Finally, DNA pellets were resuspended in ultrapure water. 

Degenerate primers for *mnp* were designed according to LC-MS/MS based protein fingerprints of MnPs [22], whereas degenerate primers for *lac* were designed based on the conserved nucleotide sequence of *lac* from related species in genus *Trametes* and are listed in Appendix A. The fragments were amplified by PCR with LA Taq DNA polymerase (Takara Bio, Kusatsu, Japan). To obtain the 5′ and 3′ flanking of the ligninolytic enzyme-encoding genes *mnp1*, *mnp2*, and *lac1*, inverse PCR was performed with the primers listed in Appendix A. Genomic DNA of *T. polyzona* KU-RNW027 was digested with restriction enzymes *Bgl*II, *Sty*I, and *BamH*I, and self-ligated using DNA ligation kit version 2.1 (Takara Bio, Kusatsu, Japan). Inverse PCR was performed with LA Taq DNA polymerase (Takara Bio, Kusatsu, Japan) using self-ligated DNA as the template and inverse PCR primer pairs (Appendix A). The inverse primers were designed based on known sequences of *mnp1*, *mnp2*, and *lac1*. All PCR products were purified and cloned into pUC19 vector. *Escherichia coli* DH5α was transformed with pUC19 recombinants and cultured on Luria–Bertani medium supplemented with 50 μg/mL ampicillin at 37 °C for 16–18 h. Recombinant plasmids were extracted using Qiagen Plasmid Purification Kit (Qiagen Inc., Germantown, MD, USA). The sequencing reaction was performed using Dye Terminator Cycle Sequencing Kit (Beckman Coulter Inc., Indianapolis, IN, USA) and M13f and M13r universal primers. Sequences were analyzed on a Beckman Coulter CEQ 2000 DNA sequencer (Beckman Coulter Inc., Indianapolis, IN, USA).

### 2.7. Cloning of cDNA of mnp1, mnp2, and lac1

Total RNA was extracted using TRIzol Reagent^®^ (Molecular Research Center Inc., Cincinnati, OH, USA). Total cDNA was synthesized from the total RNA using ReverTra Ace^®^ qPCR RT Master Mix with gDNA Remover (Toyobo, Osaka, Japan). Using the total cDNA as a template with degenerate primer pairs (MnPf and MnPr for *mnp* and Lacf and Lacr for *lac*), 0.8 kb *mnp* and 1 kb *lac* fragments were amplified by PCR with LA Taq DNA polymerase (Takara Bio, Kusatsu, Japan). Based on the nucleotide sequences of these fragments, specific primers for 5′ and 3′ rapid amplification of cDNA ends (5′/3′-RACE Kit, Roche Diagnostics, Mannheim, Germany) were designed to perform cloning of the full-length of cDNA, as shown in Appendix A. All PCR products were purified and cloned into pT7blue vector (Novagen, Madison, WI, USA). Sequences were analyzed. The cDNA sequences of ligninolytic enzyme-encoding genes were aligned with those of other fungi by Blast analysis in GenBank database (http://www.ncbi.nlm.nih.gov/). A phylogenetic tree of the deduced amino acid sequences was constructed using the Mega6 program using the maximum likelihood method with 1000 bootstrap resampling replicates [24]. A putative signal peptide was analyzed using SignalP 4.0 [25].

### 2.8. Analysis of Gene Expression by Quantitative Real-Time PCR (qRT-PCR)

Total cDNA was synthesized as mentioned in Section 2.7. The template cDNA was diluted five times with 10 mM Tris-HCl (pH 8.0). RNA was quantified by measuring its absorbency at A260 using NanoDrop One (Thermo Fisher Inc., Waltham, MA, USA). RNA purity and quality were checked by A260/280 and A260/A230 ratios and gel electrophoresis, respectively. *mnp1*, *mnp2*, and *lac1* expressions were quantified using StepOnePlus Real-Time PCR system (Thermo Fisher Scientific Inc., Waltham, MA, USA). qRT-PCR was performed using the primer sets listed in Appendix A. Reactions were conducted using Thunderbird SYBR qPCR Mix (Toyobo, Osaka, Japan) following the manufacturer’s instructions. Briefly, qRT-PCR was performed with pre-denaturation at 95 °C for 60 sec followed by 40 cycles of denaturation at 95 °C for 15 s and annealing and extension at 60 °C for 30 s. In the final step, the melting curve was analyzed. 

Standard curves of *mnp1*, *mnp2*, and *lac1* amplification were generated using PCR products amplified from total cDNA of *T. polyzona* KU-RNW027. Transcription levels are expressed as induction fold. This is a copy number of target genes, *mnp1*, *mnp2*, or *lac1*, referred to as the copy number of the housekeeping gene. In this study, the rRNA gene was used as the housekeeping gene. All experiments were performed in triplicate. 

### 2.9. Statistical Analysis

All experiments were conducted in triplicate. A pairwise multiple comparison procedure following Student–Newman–Keuls post-hoc tests was used to identify sample means that were significantly different from each other. SigmaPlot 11 (Systat Software Inc., San Jose, CA, USA) was used to calculate statistical probabilities. A *p*-value < 0.05 indicated statistically significant differences between means of maximum enzyme activities, transcription level, and fungal growth at a 95% confidence interval.

## 3. Results and Discussion

### 3.1. Ligninolytic Enzyme-Encoding Genes of T. polyzona KU-RNW027

Using MnPf and MnPr as the degenerate primers, 1014 and 1016 bp fragments of *mnp1* and *mnp2*, respectively, were obtained by PCR. Inverse PCR was then conducted to clone the 5′ and 3′ flanking sequences of these fragments. For *mnp1*, an amplified product of 1661 bp was obtained from inverse PCR. After dividing the 1661 bp into two parts at the *Bgl*II restriction site and reintegrating with the 1014 bp fragment, a 2379 bp fragment was obtained that encompassed an intact 5′ and 3′ flanking sequence of *mnp1.* In parallel, RT-PCR and 5′/3′-RACE were performed to amplify the cDNA sequence of *mnp1.* Full-length cDNA sequences of the *mnp1* gene consisted of 1331 nucleotides. Four short introns were found in *mnp1* sized between 60 and 80 bp. All intron splice junction regions belonged to the GT–AG rule [26]. The *mnp1* gene showed 86% nucleotide sequence identity with *mnp1* from *Lenzites gibbosa* CB-1 (FJ848739.2). The *mnp1* corresponded to a protein of 364 amino acids. Sequence analysis of the deduced amino acids of MnP1 by SignalP 4.0 [25] revealed the presence of a putative signal peptide of 21 amino acids, with a cleavage site between Ser-21 and Ala-22 residues. The mature protein of MnP1 contained 343 amino acids.

For *mnp2*, an amplified product of 1711 bp was obtained from inverse PCR. After dividing the 1711 bp into two parts at the *Sty*I restriction site and reintegrating with the 1016 bp fragment, a 2312 bp fragment containing intact 5′ and 3′ flanking sequence of *mnp2* was obtained. In parallel, RT-PCR and 5′/3′-RACE were performed to amplify the cDNA sequence of *mnp2.* Full-length cDNA sequences of the *mnp2* gene consisted of 1306 nucleotides. The alignment of cDNA and intact sequence *mnp2* revealed five introns. The nucleotide sequence of *mnp2* showed 84% identity with *mnp2* from *L. gibbosa* CB-1 (JQ388597.1). The *mnp2* gene predicted 366 amino-acid-encoded protein, and a putative 21 amino acid signal peptide was identified with a putative cleavage site between Gln-21 and Ala-22 residues. The mature protein of MnP2 contained 345 amino acids. 

A phylogenetic tree was constructed using the deduced amino acid sequences of MnP1 and MnP2 together with the other 34 complete amino acid sequences of class II fungal secretion heme peroxidases (Figure 1A). Phylogenetic analysis revealed that MnP1 and MnP2 of *T. polyzona* KU-RNW027 are clustered within subfamily A.2 clade. The two MnPs were short-type hybrid MnPs with no exhibited lignin peroxidase activity. They were distinctively distinguished from subclasses B and C, which are typically long-MnP and other peroxidases, respectively [27]. MnP1 is close to MnP1 from *L. gibbosa* CB-1 (ACO92620.2) at 90% identity. MnP2 has 86% identity with MnP2 from *L. gibbosa* CB-1 (AEX01145.1). MnP1 and MnP2 of *T. polyzona* KU-RNW027 had only 88% identities in their deduced amino acid sequences, suggesting that they are distinctive MnPs.

Important catalytic and conserved amino acid residues are also included in the deduced amino acid sequences of the two new MnPs and another white rot fungi MnP (Figure 2). Eight conserved cysteine residues were identified in MnP1 and MnP2. At a functional level, amino acid residues are closely related to peroxidase oxidation sites and heme pocket residues, and Mn^2+^ and Ca^2+^ binding sites [28] are conserved in the two new deduced MnP sequences.

A 1625 bp fragment of *lac1* was obtained by PCR using Lacf and Lacr as degenerate primers. To clone the 5′ and 3′ flanking of Lac1 sequence, inverse PCR was performed, yielding a 1623 bp segment. After dividing the 1523 bp into two parts at the *BamH*I restriction site and reintegrating with the 1625 bp fragment, a 3016 bp fragment of *lac1* was obtained. RT-PCR and 5′/3′-RACE were performed in parallel to amplify the cDNA sequence of *lac1.* Full-length cDNA sequences of the *lac1* gene consisted of 1809 nucleotides. The alignment of cDNA and intact sequence *lac1* revealed 10 introns. The nucleotide sequence of *lac1* has 97% identity with *lac* from *T. polyzona* MUCL:38443 (KT802746.1). The *lac1* corresponded to a protein of 520 amino acids. The deduced amino acid sequence of Lac1 showed 99% identity with laccase from *T. polyzona* MUCL:38443 (AOZ19964.1). SignalP 4.0 analysis revealed that pre-mature protein of Lac1 of *T. polyzona* KU-RNW027 consisted of a putative 21 amino acid signal peptide with a cleavage site between Ala-21 and Ala-22 residues. The mature Lac1 protein had 499 amino acids. 

Phylogenetic analysis of the deduced amino acid sequence of Lac1 was constructed with the other 35 complete amino acid sequences of multiple copper oxidases from different sources, including Basidiomycetes, Ascomycetes, plants, insects, and bacteria. The results revealed that Lac1 is clustered in the Basidiomycetes laccase clade in the same branch as laccases from the other *T. polyzona* strains (Figure 1B). 

The ligninolytic system of *T. polyzona* KU-RNW027 consisted of two MnP-encoding genes and one laccase-encoding gene. According to their deduced amino acid sequences, MnP1 and MnP2 are new manganese peroxidases in class II fungal secretion heme peroxidase, whereas Lac1 is similar to other *T. polyzona* laccases. The intact nucleotide sequences of *mnp1*, *mnp2*, and *lac1* were deposited in GenBank under accession numbers LC340363, LC464698, and LC464699, respectively, whereas full-length cDNA sequences of *mnp1*, *mnp2*, and *lac1* are LC340360, LC340361, and LC340362, respectively.

### 3.2. Promoter of Ligninolytic Enzyme-Encoding Genes

The 5′-flanking sequences upstream of the start codon ATG of the *mnp1*, *mnp2*, and *lac1* genes were obtained by inverse PCR. The putative *cis*-acting regulatory sequence elements, which are involved in their transcriptional regulations, were analyzed. The putative promoter region of *mnp1* 730 bp upstream of the start codon is presented in Figure 3A. We found one TATA box (TATAA) 79 from the start codon (ATG) in the *mnp1* promoter. No CAAT box was found in the promoter of *mnp1*, whereas two inverted CAAT motifs (ATTGG) were observed at positions –205 and –301. We found one potential light responsive element (LRE) composed of the 6 bp CCRCCC motif [20] at –307. We found two xenobiotic response element (XRE) motifs, TNGCGTG and CACGCW [29], at –120 and –607, respectively. Metal responsive element (MRE) motif, TGCRCNC [18] at –147 and –452; two copper responsive element (CRE) motifs HWHNNGCTGD [30] at –323 and –337; one iron responsive element (IRE) motif CAGTGH [31] at –533; three putative CreA-binding site (cAMP-mediated glucose repression element) SYGGRG [32] at –90, –343, and –434; and nine heat shock elements (HSEs) NGAAN [33] at –64, –230, –339, –470, –624, –669, –688, –697, and –708 were identified on the promoter region of *mnp1*.

Figure 3B depicts 894 bp of the putative promoter region of *mnp2*. One TATA box at –93 relative to start codon ATG was found in the promoter of *mnp2.* Similar to the promoter of *mnp1*, two inverted CAAT motifs were found at positions –170 and –275, and no direct CAAT box was present in the *mnp2* promoter. Remarkably, we found two LREs at –282 and –569 relative to the start codon ATG of *mnp2*. Two MREs at –139 and –408, one XRE at –195, one CRE at –298, two CreA binding sites at –78 and –652, and 10 HSEs at –54, –104, –187, –243, –318, –377, –456, –517, –558, and –808 were identified on the *mnp2* promoter region. 

We depict 906 bp of the putative promoter region of *lac1* in Figure 3C. One TATA box and two CAAT boxes were identified at –93, –203, and –844, respectively. Unlike the *mnp1* and *mnp2* promoters, only three kinds of *cis*-acting regulatory sequence elements were observed on the *lac1* promoter as MRE, XRE, and HSE motifs. Two MRE motifs were found at –389 and –802; two XRE motifs were found at –272 and –400. Ten HSE motifs were found at –231, –361, –515, –536, –646, –699, –707, –767, –826, and –852. LRE, CRE, IRE, and CreA binding sites were not identified in the *lac1* promoter.

The promoter regions of *mnp1*, *mnp2*, and *lac1* showed low sequence similarity with those of the other fungi. We found unique sequences in the respective genes. However, all promoters shared common regulatory elements, such as MRE, XRE, and HSE [19,28,30,31]. Conserved sequences, TATA boxes, and CAAT boxes were commonly found in the core promoter region as essential for gene expression. The TATA box is known as a transcription initiator site [34], whereas the CAAT box is an RNA polymerase recognition site [35]. The promoter regions of ligninolytic enzyme-encoding genes usually contain one TATA box and at least one CAAT box at positions between –30 and –1145 from the start codon to initiate the transcription [18]. In contrast, no direct CAAT box was found in the promoter regions of *mnp1* and *mnp2*. Two inverted CAAT boxes were identified in both promoters, in accordance with results reported by Chen et al. [28]. 

In previous reports, no LRE was identified on the promoters of ligninolytic enzyme-encoding genes. Interestingly, we found one and two LREs in the promoters of *T. polyzona* KU-RNW027 *mnp1* and *mnp2*, respectively. These may drive regulation by light.

### 3.3. Regulation of Ligninolytic Enzyme Production and Gene Expression by Different Wavelengths of Light

#### 3.3.1. Preliminary Plate Assay Study

Under continuous lighting, the size of the orange zone that indicates ligninolytic enzyme activities was larger than in the dark control (Figure 4). This implied that light enhances the ligninolytic enzyme production of *T. polyzona* KU-RNW027. Therefore, we determined the ligninolytic enzyme activities and their gene expressions under different wavelengths of light. However, light did not affect fungal growth, since the diameters of fungal colonies on agar plates were no different under light and dark conditions.

#### 3.3.2. Effect of Light on Transcription Levels and Enzyme Production

Light of tungsten (320–1100 nm) significantly stimulated MnP activity but not laccase activity. MnP activities were observed as being 1.5 times higher under light compared with the dark control (Figure 5A). In the dark, *T. polyzona* KU-RNW027 produced both MnP and laccase at 2.2 and 1.9 U/mL, respectively. Reduction of both enzyme activities was caused by blue light. Only 0.81 and 0.59 U/mL of MnP and laccase activities were detected under blue light, respectively. However, tungsten light and blue light did not affect fungal growth. The biomass of the fungus averaged 1.2 ± 0.07 g/L in all the conditions tested. 

The highest levels of *mnp1* and *mnp2* transcripts were observed under tungsten light irradiation at 6.5 and 3.8-fold induction of *mnp1* and *mnp2*, respectively, compared to dark condition (Figure 5B). This result showed the correlation of the induction of both *mnp* gene transcriptions by tungsten light illumination with increasing MnP activity. However, tungsten light only up-regulated the *mnp* genes but not the *lac* gene. Under blue light illumination, levels of *mnp1*, *mnp2*, and *lac1* transcripts were not significantly different from those observed under the dark control. Blue light had no effect on either *mnp* or *lac* gene expressions. However, blue light affected both enzyme activities (Figure 5A,B). 

Tungsten light up-regulated MnP activity and its gene expression compared with the dark condition. Laccase activity and its gene transcription levels were not affected by light, in good agreement with the observation that no LRE was identified on the *lac1* promoter region. Previous studies reported the effect of light on MnP, laccase, and lignin peroxidase production in submerged cultures of *C. unicolor*, *P. sanguineus*, *P. lindtneri*, and *P. chrysosporium* [14,15]. Light also stimulated the production of MnP from solid-state ash sawdust *C. unicolor* culture [36]. However, analysis of ligninolytic enzymes gene expression in *C. unicolor* revealed that wood degradation properties may not only be dependent on lighting conditions but also result from overall stimulation of fungal metabolism by daylight [37].

#### 3.3.3. Effect of Light on Ligninolytic Enzyme Stability

Compared to the equivalent dark controls, both MnP and laccase activities were relatively stable under tungsten light for the first six hours of exposure, retaining 90%–95% initial activity respectively. Thereafter, both enzymes were progressively effected to a greater extent, but still retained 60%–75% initial activity even after 72 h of exposure. Conversely, over the same time course, blue light had a much stronger negative effect, with the retained activity of both enzymes falling to 70% after only 6 h of exposure. For MnP, retained activity progressive declined to 39% after 72 h of exposure, whereas for laccase the detrimental effect became much more noticeable with increased exposure such that the retained activity of this enzyme had declined to 6% of its initial level at 72 h. (Figure 6) 

The biochemical basis of the detrimental effects of exposure to blue light on both the tested MnP and laccase from KU-RNW027 remains currently uncharacterized. However, blue light has been previously shown to effect the biological functions of many proteins [1]; specifically, blue light can change the activity of the endonuclease *Pvu*II by changing the sulfur–sulfur bond configuration of the enzyme in a way that reduces enzyme–substrate interaction [38]. Multiple cysteine residues, and therefore, potentially disulfide bonds, exist in both the tested MnP and laccase enzymes (Figure 2), and so disruption of these particular sulfur–sulfur bonds could be a possible explanation for the observed effects of blue light on these enzymes. Confirmation of the actual mode of action of blue light on KU-RNW027 requires much more extensive investigation.

## 4. Conclusions

Our findings suggested two new manganese peroxidases and one laccase in the ligninolytic system of *T. polyzona* KU-RNW027. Ligninolytic enzyme production and gene expression in *T. polyzona* KU-RNW027 can be regulated by light. This is the first report on LREs identified in the *mnp* promoter region in fungi. Tungsten light enhanced MnP production but not for laccase, whereas blue light impacted enzyme stability. Our findings effectively supported improvement of ligninolytic enzyme production in the white rot fungus *T. polyzona* KU-RNW027. This result benefits the application of an eco-friendly approach, especially in toxic aromatic compound bioremediation.

## Figures and Tables

**Figure 1 microorganisms-08-00852-f001:**
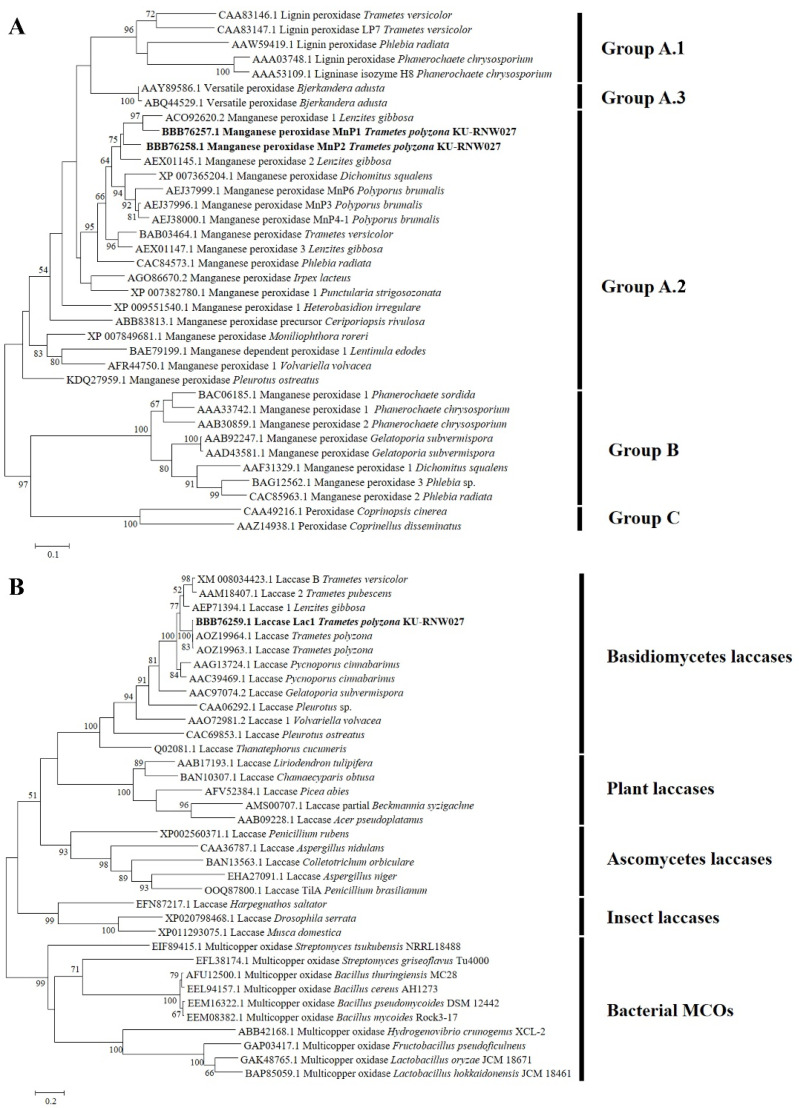
(**A**) Phylogenetic tree of class II fungal secreted heme peroxidases and (**B**) multiple copper oxidases. Maximum likelihood trees of 34 complete ORF translated amino acids for class II fungal secreted heme peroxidase and 35 complete ORF translated amino acids for multiple copper oxidases were constructed using Mega version 6.0. Confidence values above 50% obtained from 1000 bootstrap replications are indicated at branch nodes. The scale bar indicates the number of amino acid substitutions per site.

**Figure 2 microorganisms-08-00852-f002:**
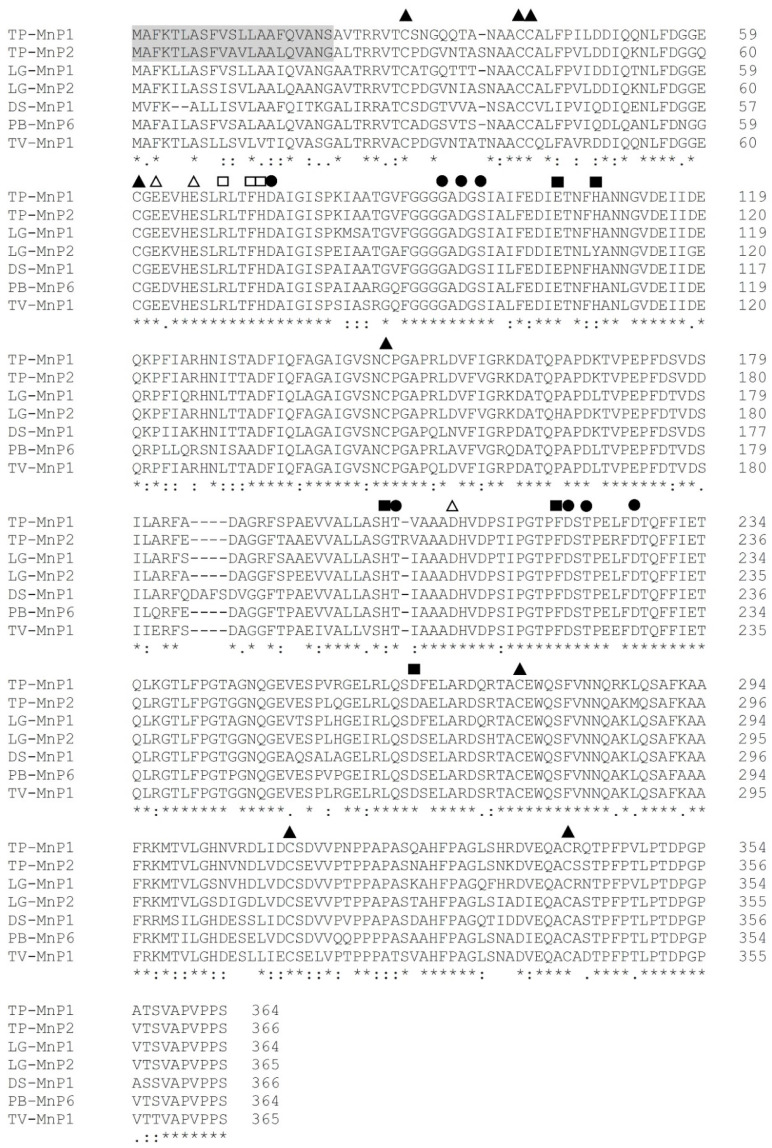
Deduced amino acid sequences of two new *mnp* genes. The shaded amino acid sequence is the predicted signal peptide. Eight conserved cysteine residues are indicated by triangles (
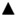
). Conserved heme pocket residues are indicated by solid rectangles (
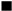
). Mn^2+^ binding sites are indicated by hollow triangles (
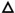
). Ca^2+^ binding sites are indicated by circles (
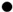
). Oxidation sites of the substrate are indicated by hollow rectangles (
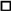
). TP: *Trametes polyzona*; LG: *Lenzites gibbosa*; DS: *Dichomitus squalens*; PB: *Polyporus brumalis*; TV: *Trametes versicolor*. A putative signal peptide was analyzed using SignalP 4.0 [25].

**Figure 3 microorganisms-08-00852-f003:**
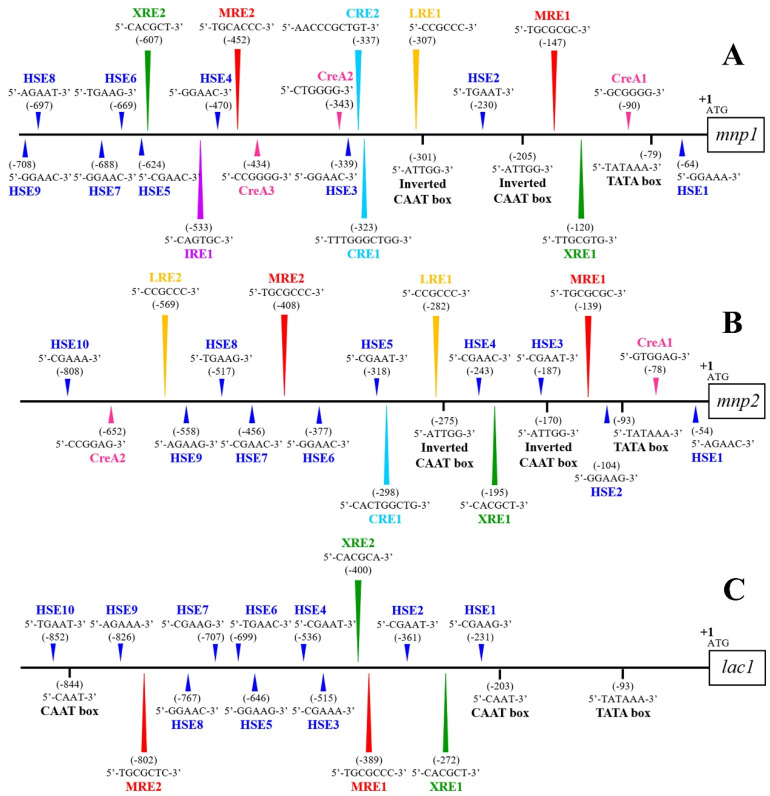
The scheme of the *cis*-acting element of (**A**) *mnp1*, (**B**) *mnp2*, and (**C**) *lac1* upstream promoter from ATG. LRE, light responsive element; MRE, metal ion responsive element; XRE, xenobiotic responsive element; CRE, copper responsive element; IRE, iron responsive element; CreA, CreA binding site or cAMP responsive element; and HSE, heat shock element. The first nucleotide of start codon (ATG) is labeled as +1.

**Figure 4 microorganisms-08-00852-f004:**
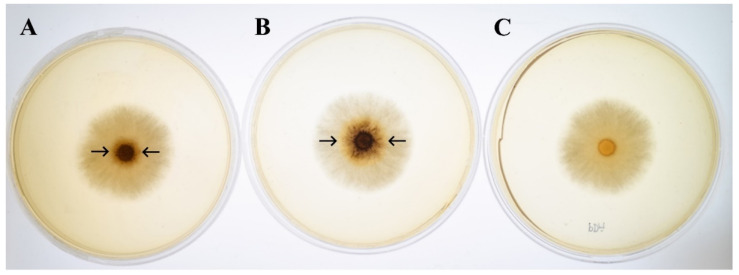
Ligninolytic enzyme activity of *T. polyzona* KU-RNW027 on agar plates. The fungus was grown on PDA supplemented with 1 mM 2,6-DMP under (**A**) dark and (**B**) continuous lighting conditions. The fungus grown on PDA without 2,6-DMP under light was used as a control (**C**). Ligninolytic enzymes oxidized 2,6-DMP to produce the orange product of the benzoquinone compound. Arrows indicate visualization of the enzyme activity.

**Figure 5 microorganisms-08-00852-f005:**
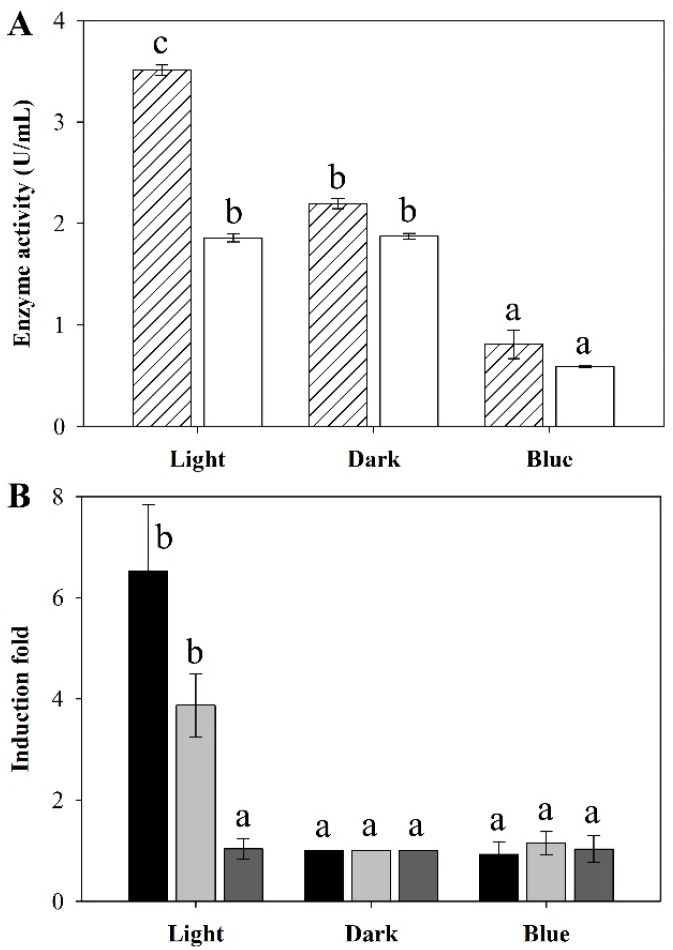
Effect of different wavelengths of light on (**A**) ligninolytic enzyme productions and (**B**) gene expressions in *T. polyzona* KU-RNW027. MnP (
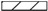
) and laccase (
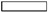
) activities in the culture supernatant were determined after 7 days of cultivation under different wavelengths of light. Transcriptional levels of the *mnp1* (
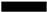
), *mnp2* (
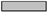
), and *lac1* (
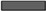
) genes were determined using mycelia harvested from the cultures after 7 days of cultivation under different wavelengths of light. Transcription levels of *mnp1*, *mnp2*, and *lac1* are shown as induction fold values relative to those observed under the dark condition. Error bars represent standard deviation (*n* = 3). a, b, and c represent significant enzyme production and gene expression compared to the control (*p* < 0.05).

**Figure 6 microorganisms-08-00852-f006:**
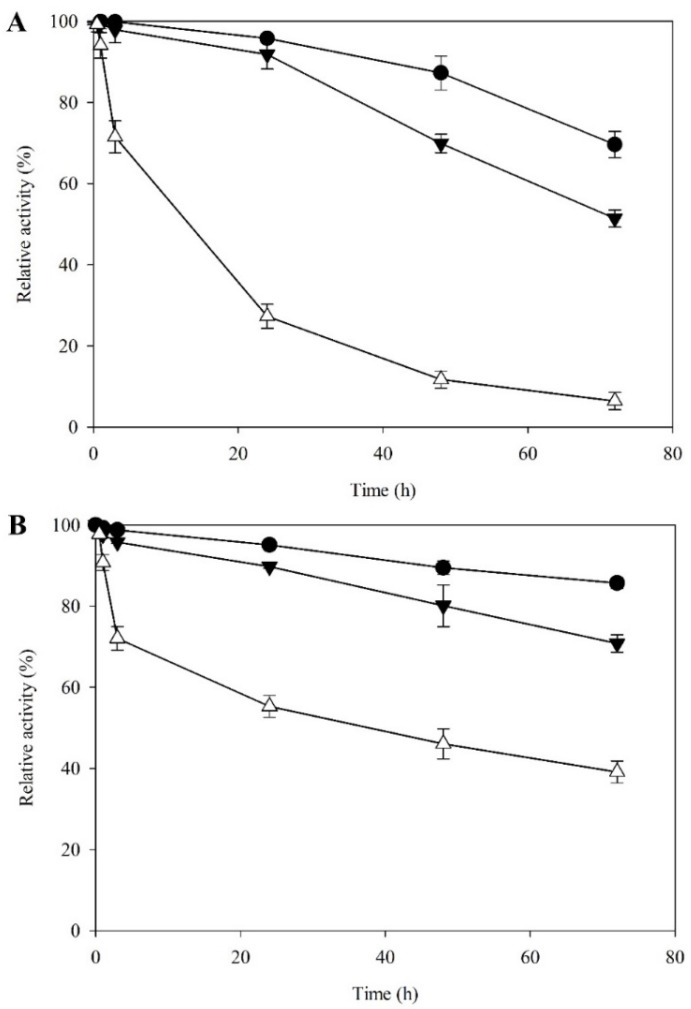
Time course of (**A**) laccase and (**B**) MnP stability under different wavelengths of light. Error bars represent standard deviation (*n* = 3). Dark condition (
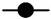
), tungsten light (
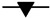
), and blue light (
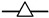
).

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
