# Peer review of "Light Regulation of Two New Manganese Peroxidase-Encoding Genes in *Trametes polyzona* KU-RNW027"

_microorganisms, 2020, doi:10.3390/microorganisms8060852_

Round 1

Reviewer 1 Report

Your manuscript (Ms.) focus on the occurrence of three ligninolytic enzyme-encoding genes on a strain Trametes polyzona KU-RNW027, and the two MnPs are new enzymes belonging to the class II. Also, the light effects were tested in fungal growth, enzymes activities and enzymes genes. Your work is important since added knowledge and the findings could have an impact in industrial production of these enzymes.

Nevertheless, the Ms. could be improve in order to facilitate its reading and comprehension. Please find bellow some suggestions/advices in that pursue.

a) Abstract – needs a brief introduction to the theme, objectives and to the experiment. With the exception of the first phrase, the abstract is only results.

b) Introduction – needs, in the end, the objectives of the work clearly stated.

c) 2.1. Microorganism and inoculum preparation – this point is very confuse and important information is missed. Separate the fungal current maintenance (in PDA) from long term preservation (-20 ºC, in glycerol 20%, using pellets?), from the experiment itself (the fungus grew in Kirk’s broth – conditions, time of the experiment?)

d) 2.2. Cloning of ligninolytic enzyme-encoding genes and promoter region – Although the citation, briefly explain the genomic DNA extraction.

e) In Material and Methods (M&M), the work flow must be changed. Suggestion, by order: 2.1., 2.4., 2.5., 2.6., 2.7 (briefly describe the enzymatic activities determination), 2.2, 2.3, 2.8., 2.9

f) Results

Lines 152-157 – “All intron splice junction regions belonged to the GT-AG rule [23] (…) [26]”. Presumably numbers 23 and 26 are references that should be removed from this section and transferred to discussion. Same observation to lines 171-172; 226-233; (Transfer all these paragraph to discussion, or merge results and discussion in one section – I think ii would be the best option).

If you used the by SignalP 4.0 software, it must be referred in M&M section (data analysis).

Fig. 1 – explain the reference behind the protein (ACO92620.2) in figure’ legend;

Fig. 2 – using the SignalP? Complete the fig. legend

Fig. 3 – explain letters above the bars (statistical significance, probably?)

In text, whenever reference to enzyme activities, transcription level or fungal indicate the significance (p>0.05; …)

Considerer remove paragraph 290-292, and use it in discussion or conclusion.

I recommend the merge of “Results” and “discussion” sections in order to enhance the communication between authors and future readers.

Reviewer 2 Report

As the plant biomass is chemically complex, the organisms, capable to degrade it, developed various strategies to successfully decompose polysaccharides and to cope with lignin. Among great number of species, fungi are very interesting group and brought the researchers attention in recent years. Moreover, the newest studies proved that synthesis of these enzyme is regulated by light beside of number of chemical and physical factors. However, still little is known about the details of molecular mechanisms underlying the enzymes production and their secretion in fungal organisms. There is lack of papers dealing with wood degradation relying on lightning conditions. Consequently, the presented paper is of high importance as it gives the best possible knowledge about it. However, in the opinion of the reviewer the paper needs some corrections listed below:

Major:

  1. The genome of Trametes polyzona is available in JGI website. Even if the strains are different, was it necessary to perform sequencing of genes coding for MnP and laccase in the presented paper?
  2. Based on the genome sequence would it be possible to give the number of polyzona photoreceptors?
  3. Are there any LRE in MnP promoter sequence in mentioned genome? Are there any other differences?

Minor:

  1. Abstract – “This is the first time…..been investigated” – Not the first time. Please see Pawlik et al. (2019) International Journal of Molecular Sciences 20(2)290
  2. Introduction – “A few studies….” – Not so few see above and Pawlik et al. (2019) Acta Biochimica Polonica 66(4) 419-425.
  3. Introduction, line 52 – “However, ….genes.” – the same sentence is in the line 44.
  4. Material and methods, line 101 and 107 – please give more details on white light. Why different sources of blue and white light was used? 500 lux wasn’t too intensive lightning conditions? Trametes is a wood species living in shadows with green light as dominant color?
  5. Material and methods, line 119 – “…[24]…[25]..” – Please provide citation as “…name [24]…name [25]..”. And please give original paper citing assays. According to paper of Teerapatsakul laccase was assayed with ABTS not DMP. If DMP was used in enzyme mixture how activity of laccase was distinguished from MnP activity?
  6. Material and methods, line 121 – “….denatured enzyme…” – how denaturation was performed? Was it completely dead?
  7. RT-PCR section – Could you describe the dilutions of templates? What method was used for RNA quantification? Any negative control was used? Could you give the details on PCR amplification temperatures and time?
  8. Results, line 143 – if you blast polyzona genome with any of your MnP sequence how many hits will you get? What will be the query coverage etc?
  9. Results, line 219 – did you submitted to GenBank the promoter sequences?
  10. Figure 3 – in my opinion it would be better to present this data as table.
  11. Figure 4 – perhaps a variant of fungus grown in dark without DMP would be helpful.
  12. Discussion – As white light is composed of blue light among others how the authors would explain their results? Perhaps it is not that blue light inhibited the enzyme activity but the absence of red and green was the reason?

Author Response

This manuscript is a resubmission of an earlier submission. The following is a list of the peer review reports and author responses from that submission.